# The Current Knowledge on the Pathogenesis of Tissue and Medical Device-Related Biofilm Infections

**DOI:** 10.3390/microorganisms10071259

**Published:** 2022-06-21

**Authors:** Enea Gino Di Domenico, Alessandra Oliva, María Guembe

**Affiliations:** 1Department of Biology and Biotechnology “C. Darwin”, Sapienza University of Rome, 00185 Rome, Italy; enea.didomenico@uniroma1.it; 2Department of Public Health and Infectious Diseases, Sapienza University of Rome, 00185 Rome, Italy; alessandra.oliva@uniroma1.it; 3Department of Clinical Microbiology and Infectious Diseases, Hospital General Universitario Gregorio Marañón, 28007 Madrid, Spain; 4Instituto de Investigación Sanitaria Gregorio Marañón, 28007 Madrid, Spain

**Keywords:** biofilm, pathogenesis, tissues, devices, clinical impact

## Abstract

Biofilm is the trigger for the majority of infections caused by the ability of microorganisms to adhere to tissues and medical devices. Microbial cells embedded in the biofilm matrix are highly tolerant to antimicrobials and escape the host immune system. Thus, the refractory nature of biofilm-related infections (BRIs) still represents a great challenge for physicians and is a serious health threat worldwide. Despite its importance, the microbiological diagnosis of a BRI is still difficult and not routinely assessed in clinical microbiology. Moreover, biofilm bacteria are up to 100–1000 times less susceptible to antibiotics than their planktonic counterpart. Consequently, conventional antibiograms might not be representative of the bacterial drug susceptibility in vivo. The timely recognition of a BRI is a crucial step to directing the most appropriate biofilm-targeted antimicrobial strategy.

## 1. Introduction

Biofilms are the most relevant driver of persistent infections and a major healthcare problem. Biofilm formation occurs on various body surfaces, including the skin or mucosal surfaces of the respiratory and digestive tract or medical devices such as catheters, contact lenses, heart valves, and prostheses [1]. Aggressive and long-term antibiotic therapies, based on drug resistance profiles, demonstrated poor efficacy in eradicating biofilm infections since a minimum bactericidal concentration cannot be achieved in vivo without posing serious risks of adverse effects to the host [2]. Moreover, in biofilm-related infections (BRIs), planktonic cells can disseminate from the primary site of infection and spread into the bloodstream, raising the risk of disease and death, particularly in immunocompromised patients [3,4,5]. In most cases, removing the infected device associated with surgical debridement implant replacement and targeted antibiotic therapy is the only efficient way to eradicate a BRI [6,7]. Indeed, biofilms exhibit broad and intrinsic multidrug tolerance to environmental and chemical agents, allowing microbial cells to survive a transient exposure to antibiotics without developing resistance [8]. In contrast to the genetically acquired antibiotic resistance, antibiotic tolerance is caused by a variety of mechanisms, such as a decreased drug penetration within the extracellular matrix, the enzymatic inactivation of the antimicrobials, slow-growing or non-dividing persister cells, an altered chemical microenvironment, and adaptive stress responses [9]. The systemic administration of antibiotics based on the antimicrobial susceptibility profiles can eliminate planktonic microorganisms released from the biofilm matrix but is often ineffective in treating BRIs [8]. Consequently, biofilm-growing microbial cells can persist in the host if not promptly eradicated during the acute infection phase. The matrix, which is formed from a complex array of extracellular polymeric substances (EPS), confers unique attributes of the biofilm lifestyle and enhanced drug tolerance [10]. The EPS mainly consists of exopolysaccharides, extracellular nucleic acids (eDNA and eRNA), proteins, lipids, and other biomolecules mediating biofilm formation and architecture in a dynamic process [10,11]. For example, in biofilms, the curli protein, together with cellulose, contributes to the desiccation tolerance of the biofilm [12]. Moreover, eDNA, which provides surface adhesion and structural integrity to the matrix, induces antibiotic tolerance through different mechanisms [13,14,15,16]. Therefore, the clinical implications of this finding suggest that antibiotic efficacy may be increased by weakening the biofilm structure. Moreover, an early and aggressive antibiotic therapy, including biofilm-targeted antimicrobials is highly recommended for effective biofilm eradication [7]. Unfortunately, the current methods to evaluate microbial biofilm are usually time-consuming, costly, and hardly applicable in routine diagnostics [7]. Thus, the diagnosis of a biofilm-associated infection still represents an area of serious concern for the clinical management of patients. The timely recognition of a high biofilm producer before developing a mature biofilm matrix may provide key decision-making elements for the most appropriate medical or surgical intervention. Moreover, the assessment of biofilm-induced antibiotic tolerance, which circumvents conventional drug resistance mechanisms, may provide novel insights into the therapeutics and prevention strategies against biofilm-related infections (type, doses, and duration of antimicrobial therapy). This discussion focuses on the pathogenic role of BRIs in the management and clinical outcome of medical-device-related infections as well as biofilm-associated infections highly tolerant to systemic antimicrobial therapy.

## 2. Biofilm and Chronic Infections in Tissues

Although biofilm formation has been commonly associated with the presence of foreign bodies, it can also develop on human tissues, leading to either localized infections or acting as a reservoir for pathogens preceding clinical infection with planktonic bacteria [17].

Tissue-related biofilm infections are often chronic and characterized by (i) a significantly increased tolerance to antibiotics, (ii) a high capacity to evade the host immune defenses, and (iii) an inflammatory response further promoting tissue damage [18,19].

Several infections may be recognized as being characterized by biofilm formation and include, amongst others, infective endocarditis (IE), osteomyelitis, chronic nonhealing wounds, and bacterial persistence in the lungs of patients with cystic fibrosis (CF) or chronic obstructive pulmonary disease (COPD) [20] (Figure 1).

### 2.1. Pathogenesis

Along with the recognition that an increasing number of diseases are associated with biofilm formation, the research on the diagnosis, pathogenesis, and treatment of these infections has evolved, too. As a general concept, biofilm infections are chronic in nature since biofilm-embedded microorganisms are in a stationary growth phase and are protected by both the host immune system and antibiotics [21].

With regard to tissue biofilm diseases’ pathogenesis, whether the biofilm itself is the cause of the disease or whether biofilm-embedded bacteria exploit a favorable environment for colonization caused by the underlying disease is still a matter of debate.

In this context, several characteristics of biofilm are implicated in the pathogenesis of diseases. For instance, the presence of biofilm within living cells plays an important role in recurrent osteomyelitis [31]. Furthermore, persistent microorganisms within biofilms may stimulate a local inflammatory response that can cause or aggravate tissue damage, as happens in CF. The complex structure of biofilm represents an ideal environment for attracting different bacterial species, thus explaining the not negligible percentage of polymicrobial biofilms, as has been observed in chronic nonhealing wounds. Biofilm can also be linked to the severity and prognosis of disease, as in CF. Finally, biofilm may act as a pathogen reservoir preceding the progression towards an overt infection caused by planktonic bacteria, as observed for uropathogenic *Escherichia coli* in the urinary bladder [21].

From a therapeutic point of view, biofilm-associated infections are often treated with a combined surgical and medical approach. In fact, the surgical debridement of the infected tissue or the removal of the implant is mandatory in the majority of cases, since medical therapy only, even with antimicrobials exhibiting bactericidal activity, is not sufficient to eradicate biofilm-embedded infections [32].

This explains the risk of recurrence at therapy interruption, even after decades. For instance, in the presence of osteomyelitis, antimicrobial therapy is able to reduce the bacterial burden to a minimum; however, vital bacteria may persist within osteoblasts or bone sequestra, leading to a subsequent infection recurrence (i.e., fistula formation, persistent pain) [31].

### 2.2. Infective Endocarditis

IE is associated with high morbidity and mortality [22]. Staphylococci and streptococci account for approximately 80% of cases, with *Enterococcus* spp. as the third leading causative agent of IE [22]. From a historical point of view, IE has been recognized to be a biofilm infection caused by the viridans group streptococci by Costerton et al. in 1999 [33], while Marrie et al., in 1987, identified bacterial colonies within a matrix material on the valves of IE patients using electron microscopy [34].

Indeed, in the vegetation, which is the typical lesion of IE, biofilm bacteria are embedded within a matrix composed of extracellular material, fibrin, and platelets [35].

The process of vegetation formation includes several passages: first, the endothelial surface of the valve is damaged; then, a sterile lesion made of platelets and fibrin is formed; afterwards, bacteria present in the blood during a bacteremic episode start to adhere to the thrombus until a mature biofilm is formed. This mature biofilm represents a real sanctuary protecting infecting microorganisms from both the immune system and the antibiotics [21].

The presence of vegetation may cause valve dysfunction by promoting local bacterial invasion, destroying valve cusps and leaflets, and extravascular invasion into surrounding tissues. Furthermore, the seeding of planktonic cells from the vegetation into the bloodstream is responsible for bacteremia, sepsis, and, to a lesser extent, septic embolization. Although bacteremia may be controlled by antimicrobials and the patient’s immune system, persister cells in the deeper layers of the biofilm vegetation can survive despite multiple antibiotic treatments and, therefore, surgical treatment may be required for the clinical cure of IE [22,33,35].

### 2.3. Chronic Osteomyelitis

Chronic osteomyelitis is a difficult-to-treat bone infection associated with high morbidity and economic burden, resulting from either endogenous (hematogenous osteomyelitis) or exogenous seeding. Staphylococci account for approximately 75% of cases, with *S. aureus* being the most common pathogen [23,24], whereas anaerobes and Gram-negative bacilli, such as *Pseudomonas* spp. and Enterobacterales, are more frequently reported in polymicrobial osteomyelitis [25].

Chronic osteomyelitis is characterized by extensive devascularized bone, low-grade inflammation, and the presence of bacterial biofilm, which is considered one of the leading causes of failure in microbiological eradication [36]. Furthermore, bacterial persistence deep within bone is responsible for the observed high recurrence rates, even after a long period of time [31]. This is especially true for biofilm-growing *S. aureus*, which can remain quiescent for several years until it causes the infection [31].

### 2.4. Chronic Nonhealing Wounds

A chronic wound is characterized by an interruption in the wound healing process. Historically, wound infections were considered to be caused by planktonic bacteria; nevertheless, more recently, it was shown that chronic wounds represent biofilm infections instead, and that biofilm formation is the reason why wounds or ulcers (i.e., diabetic foot) may become chronically infected.

As a matter of fact, recently, an international consensus for clinical indicators of possible biofilm in a wound was published, including, amongst others, the failure or recalcitrance to appropriate antibiotic treatment, the recurrence of delayed healing on the cessation of antibiotic treatment, increased exudate/moisture, a low level of chronic inflammation, and delayed healing in spite of optimal wound management [37].

These conditions represent the ideal milieu for biofilm formation, which accounts for almost 90% of chronic wounds. In turn, bacterial biofilms may promote localized tissue hypoxia and reduce the oxygen availability required for the normal healing process [38].

Furthermore, bacteria within the biofilm may significantly contribute to the inflammation of the wound, as described for rhamnolipids produced by *P. aeruginosa* [26].

Chronic wound biofilm infections are often polymicrobial, and the number of species is frequently underestimated due to the difficulties in culturing methods, with significant consequences in terms of the appropriate therapeutic management. For example, a recent study evaluating the composition of the bacterial communities present in chronic nonhealing wounds found that Staphylococci and *Pseudomonas* spp. accounted for 63 and 25% of all wounds, respectively, followed by anaerobic bacteria and bacteria traditionally considered commensal [27].

### 2.5. Cystic Fibrosis

Lung disease is the primary cause of morbidity and mortality in patients with CF [39]. Biofilm is considered part of the etiology of CF, and its formation is also involved in disease progression due to biofilm-induced chronic inflammation and subsequent pulmonary exacerbations, tissue damage, progressive decline of lung function, and, in the end, the need for an organ transplant [39].

Indeed, in response to the presence of biofilm, polymorphonuclear leukocytes infiltrate the area, producing chronic inflammation with subsequent tissue damage, loss of lung function, and obstruction of the airways. Furthermore, the metabolic activity of bacteria and cells favors the development of anaerobic conditions by consuming available oxygen [40], which, in turn, seems to favor the biofilm of *P. aeruginosa* [41].

In this context, biofilm-forming *P. aeruginosa* and *S. aureus* are the most involved pathogens, although *Burkholderia* spp., *Stenotrophomonas maltophilia*, and *Achromobacter* spp. are also commonly found in CF lungs [28].

*P. aeruginosa* is present in the sputum of CF patients either in mucoid, large colonies or in non-mucoid, small colonies. The mucoid variant is characterized by a high production of alginate and is found in patients with chronic biofilm infection; alginate from mucoid colonies is, therefore, considered a biofilm-specific antigen [7].

Similar to what has been observed in other biofilm infections, antibiotic tolerance is recognized as the main reason for both bacterial persistence in CF lungs and treatment failure [42]. Furthermore, in vitro results of conventional antimicrobial susceptibility testing often fail to predict the in vivo response to the antimicrobial agents and, consequently, the clinical outcome [7,32]. Indeed, antibiotic therapies based on conventional planktonic antimicrobial susceptibility testing may not be effective for biofilm-embedded bacteria. This in vitro/in vivo discrepancy may explain their poor clinical efficacy in eradicating the infections and the frequent rate of pulmonary exacerbations during the course of the disease.

Although the disease pathogenesis is different from CF, and older patients are typically affected, subjects with COPD or bronchiectasis experience bacterial persistence and biofilm-related airways infections and exacerbations, with *P. aeruginosa* and *S. aureus* representing the most prevalent involved pathogens [43].

The microbial communities in the lower respiratory tract of patients suffering from chronic respiratory diseases such as CF, COPD, and/or bronchiectasis change as the underlying disease evolves, often becoming less diverse and dysbiotic and correlating with infection recurrences and the worsening of patients’ clinical conditions [32].

### 2.6. Dental Infections

The oral cavity comprises more than 700 different bacterial species, with saliva containing up to 10^8^–10^9^ bacteria per milliliter [44]. Oral bacteria adhere to the teeth and initiate the formation of a dental biofilm, which causes diseases in the teeth and their supporting tissues, resulting in dental caries and periodontal diseases, respectively [29].

Only about 50% of the dental biofilm microbiota can be identified by the use of traditional culture methods [45]. Along with culture-independent molecular biological methods, the principal genera constituting the oral biofilm are found to be *Streptococcus*, *Veillonella*, *Granulicatella*, *Neisseria*, *Haemophilus*, *Corynebacterium*, *Rothia*, *Actinomyces*, *Prevotella*, *Capnocytophaga*, *Porphyromonas*, and *Fusobacterium*.

However, the composition of the dental biofilm varies not only among different sites in the oral cavity but also among individuals [29].

Dental biofilms are part of the resident oral microflora, which is balanced and considered beneficial to the host. However, if this homeostasis is altered, i.e., as a consequence of poor oral hygiene or drugs, a shift in the microflora may occur by allowing more virulent bacterial species to become dominant and, thus, favoring the development of dental diseases [29].

Dental biofilm is first established in between adjacent teeth and along the gingival margin, forming the supragingival biofilm; this may gradually spread along the root of the tooth into the periodontal pocket, forming a subgingival biofilm. The corresponding diseases are dental caries and periodontal diseases, respectively.

It is worth noting that bacteria from dental biofilm may spread into the systemic circulation and, therefore, cause bacteremia, as a consequence of either dental treatment causing bleeding (i.e., tooth extractions, oral surgery) or even during daily habits (i.e., tooth brushing, chewing). In the majority of cases, the resulting bacteremia is transient and does not cause infections; however, in some circumstances, bacteremia from the oral cavity may be the cause of IE [46]. In fact, oral streptococci are amongst the most commonly reported causative agents of IE [47].

Therefore, controlling dental biofilms by means of proper oral hygiene and regular dental care is crucial to prevent either local or systemic disease development. When an oral disease has been established, the removal of deep biofilm by endodontic treatment or surgical intervention is necessary.

## 3. Biofilm and Device-Related Infections

As with tissues, biofilm can also adhere to foreign bodies, which corresponds to devices implanted inside the body or forming a connection between an inner or outer body surface where normal microbial flora is present in a sterile anatomical compartment inside the body [7]. There are many clinical situations where biofilm-related device infections (BRDIs) can occur (Figure 2). However, we will focus on the most relevant ones (Figure 3).

BRDIs are a considerable healthcare burden and account for approximately 65% of all healthcare bacterial infections [48,49]. In general, the pathogenesis of BRDI is common for all sources, but it may vary depending on factors such as the device material or duration [50,51,52]. In the first step (attachment), bacteria (or yeast) adhere to the solid surface, colonize it, and they start to upregulate a group of genes that allow the cells to be tolerant to antibiotics while producing the extracellular matrix (EPS), which is made up of polymeric sugars, extracellular DNA, bacterial secreted proteins, host components (mainly fibrin), and many other substances (aggregation) [53]. From within the matrix, bacterial cells secrete quorum sensing molecules, which direct the gene expression of the bacteria within the community (maturation) [54]. Then, some cells can detach from the biofilm, and microbial dispersal occurs into the surrounding tissue cells or the bloodstream (detachment).

### 3.1. Catheter-Related Bloodstream Infections (C-RBSI)

C-RBSI is one of the most significant nosocomial infections. Its incidence can be highly variable depending on several factors, such as the type of catheter, patient underlying disease, medical ward, site of insertion, type of material, etc. It has been estimated that between 15% and 30% of all nosocomial bacteremia are catheter-related. Generally, incidence rates range around 1–2 C-RBSI episodes/1000 admissions (or catheter days in ICUs) [63,64].

Regarding the etiology, gram-positive bacteria are the most common causative agents (especially coagulase-negative Staphylococci). However, it may also be variable depending on the type of catheter or the patient’s comorbidities, i.e., hemodialysis catheters are more frequently associated with gram-negative bacilli and peripheral catheters to *S. aureus* [55].

Biofilm formation on the catheter surface can be due to two main routes: during insertion (extraluminal) through poor skin disinfection of the patient or poor hand hygiene of the health care worker, or during maintenance (intraluminal), where catheter hubs are contaminated by an inadequate manipulation, resulting in colonization. This colonization can occur within the first 24 h after insertion (extraluminal) and up to 7–14 days (intraluminal). Once the biofilm is established, it can be identified within 48 to 72 h, so at this moment, the effectiveness of the immune system’s phagocytes and complement system decreases, and antibiotic susceptibility decreases around 1000-fold, causing further difficulty in eradication [55].

Once formed, biofilm can be difficult to treat without removing the catheter, so the prevention of biofilm formation is mandatory [65,66]. However, more progress is being made to find an effective lock therapy to eradicate the biofilm without having to remove the catheter [67,68,69,70].

### 3.2. Ventilator-Associated Pneumonia (VAP)

The development of biofilm on the surface (both internal and external) of endotracheal tubes (ETTs) is related to the development of VAP, which occurs in 9–27% of all intubated patients [71]. It occurs because of the impairment of host-defense mechanisms and the introduction of bacteria in the sterile airways, which favor biofilm development on the distal part of the ETT. The entry of biofilm pieces and cells into the sterile lungs can lead to infection and VAP [72,73].

*Pseudomonas aeruginosa* is one of the predominant microorganisms causing VAP, followed by *S. aureus* and *Escherichia coli* [56]. However, etiology may be different among ICU wards depending on the local epidemiology.

Regarding optimizing diagnostic techniques, the main goal is to find new strategies capable of detecting the presence of biofilm without removing the ETT when it is necessary to maintain. For example, catheter-based optical coherence tomography has been demonstrated to be a successful tool for detecting in vivo ETTs’ biofilms in intubated critical care patients [74].

As ETTs are difficult to remove when VAP appears, preventive measures are required to reduce biofilm formation. A recent randomized controlled trial demonstrated that noble metal alloy ETTs reduce the incidence of VAP, ventilation days, and ICU stays for patients in mechanical ventilation [75]. Although it must be assessed in a clinical setting, the application of a selective digestive decontamination solution used as “lock therapy” in the subglottic space seems to be a promising prophylactic approach that could be used in combination with the oro-digestive decontamination procedure in the prevention of VAP [76].

### 3.3. Prosthetic Joint Infections (PJIs)

Implant-related infection occurs following approximately 5% of all elective and emergency orthopedic procedures [77].

The highest risk period for developing a PJI is during the first 2 years after the arthroplasty procedure, while soft tissue healing and postoperative inflammation are still present. Surgical and patient factors may be directly associated with biofilm formation on the prosthetic joint [57]. In particular, PJIs are classified as early, delayed, or late. Early infection occurs within <3 months, delayed between 3 and 12 months, and late >12 months after implantation. Early and delayed infections are mainly caused by microorganisms that migrate at surgery, while late infections are probably acquired hematogenously [78].

The most frequent causative agents are Staphylococci, followed by gram-negative bacilli, polymicrobial infection, and anaerobic infection. However, as in other infections, the etiology may be influenced by factors such as the time from surgical intervention (as explained above, i.e., the three stages of PJI), soft tissue complications, concomitant or preceding infection, previous colonization, and the patient’s comorbidities [57].

During microorganisms’ adhesion, there are certain features of medical implants that favor biofilm formation. For example, polymer surfaces act as reactive sites that are more susceptible to microbial adherence than stainless steel and titanium, which are directly colonized by tissue cells, providing a protective effect against bacterial pathogen colonization [78].

Although sonication has been demonstrated to be the most suitable technique for detecting biofilm on the prosthesis surface [7], the image-guided arthroscopic approach provided an accurate and direct visualization of the bacterial biofilms on prosthetic joints [79].

Regarding preventive approaches to PJI, promising new strategies have been described, such as a fast-resorbable antibiotic-loaded hydrogel implant coating, which reduces post-surgical site infections after internal osteosynthesis for closed fractures [80]. In addition, cement bone containing gentamicin and rifampicin microcapsules has demonstrated superiority versus the isolated use of gentamicin in treating PJI in a rabbit model [81,82].

### 3.4. Prosthetic Heart Valve Infection

Infections related to electrostimulation devices may involve the generator, the electrodes, or both and can trigger endocarditis. The incidence of infection associated with vascular grafts is around 6%, with a mortality rate of 15–50%. Staphylococci cause 60–80% of these infections, but the viridans group streptococci and *S. bovis*, *S. pneumoniae*, beta-hemolytic streptococci (mainly *Streptococcus agalactiae*), *Abiotrophia* spp., and *Granulicatella* spp. may also be involved [58]. However, depending on the etiology, the material surface of the prosthesis can affect its adhesion, as *S. epidermidis* and *P. aeruginosa* are dependent on pyrolytic carbon surface free energy and roughness, whereas *S. aureus* adhesion appears to be independent of these factors [83].

The definite diagnosis of prosthetic valve endocarditis can only be certainly established by histological and microbiological examination of the vegetations. Molecular techniques on tissues are also useful and serology is essential for the diagnosis of difficult-to-grow pathogens. Cultures of the pocket and wires after the removal of the device are useful in the identification of the microorganism. Percutaneous puncture–aspiration of the device pocket is not recommended [84].

Recently, a novel in vitro model for bacterial biofilm growth on porcine aortic roots was developed to mimic infective endocarditis, which may serve as a baseline for further research on therapy and prevention [85].

Moreover, updates on the preventive and therapeutic strategies are being explored to eradicate staphylococcal biofilm formation and related infections. As cell surface dynamics is an important determining factor in the attachment of staphylococcal to biomaterial surfaces, modifications based on the nanopatterning and conjugation of biomaterials with antibiotics are promising preventive approaches [86].

### 3.5. Breast Implant Infections (BII)

In breast surgery with implants, infection of the surgical site as a consequence of bacterial colonization of the implant is one of the most relevant complications [87]. The presence of bacterial biofilm has been correlated with the development of capsular contracture, which is the most frequent complication in breast implant surgery reported in 5.2% to 30% of patients with breast implants [59,88]. So, for a proper diagnosis of BII, cultures of breast implant and contiguous material are required [7,84,89]. The most frequent colonizing microorganisms in breast implant cultures are Staphylococci (*S. epidermidis* and *S. aureus*) and anaerobes [59,60,61].

One of the most recent studies regarding preventive measures during breast surgery demonstrated that povidone–iodine used for implant irrigation was the best antiseptic capable of eradicating biofilm [87,90]. However, povidone–iodine absorption may be directly related to implant texture rather than to exposure time [87]. Moreover, Barnea et al. demonstrated that pre-treating implants with plasma before immersion in the povidone–iodine solution for only 5 s significantly improved its absorption [91].

### 3.6. Contact Lens Infection

Contact lenses (CLs) are one of the market’s most commonly used medical devices, worn by more than 140 million people worldwide. There are two main classes of CLs: rigid and soft lenses. The two types differ in their composition and susceptibility to microbial infections [92]. Soft lenses are more common than rigid lenses but are also at higher risk for microbial infections [92]. Surface colonization by different bacteria and fungi on CLs has been identified as a significant risk factor for eye pathologies, such as giant papillary conjunctivitis and keratitis [93,94]. In particular, microbial biofilms on lens surfaces have been associated with keratitis in humans and animal models [95,96]. Bacterial keratitis is a frequent contact lens complication that may result in corneal infection [62,97]. In particular, bacterial keratitis increased in prevalence after the introduction of soft lenses in the 1970s [98]. *P. aeruginosa* and staphylococci are the most common causative agents in CL infections and are well-known for colonizing the corneal surface as biofilms [62]. Fluoroquinolones are frequently recommended as the first-line antibiotic in treating CL-related infections [99].

However, increasing fluoroquinolones resistance among ocular isolates was described in both methicillin-susceptible (MSSA) and methicillin-resistant (MRSA) *S. aureus* strains [100,101]. Antibiotic resistance for *P. aeruginosa* ocular isolates varies, with multidrug resistant (MDR) strains ranging from 6.52 to 42.9% [102,103]. A growing interest in ocular biofilms over the past decade has allowed for the development of in vivo cornea models and improved the methods of imaging biofilm bacteria [62]. Biofilm formation has been observed at the corneal surface during experimental infection, and at the same time, bacteria obtained from corneal infections were able to form a biofilm [104,105,106]. In vivo infection models have also been developed to study bacterial keratitis caused by biofilm growing cells on contact lenses in rabbits and mice [107,108]. More recently, ocular bacterial communities have been studied using culture-independent methods with 16S rRNA gene sequencing in healthy subjects and in people with eye diseases [109,110,111]. These studies detected additional bacteria potentially involved in pathological conditions, such as *Acinetobacter*, *Aquabacterium*, *Bacillus*, *Bradyrhizobium*, *Brevundimonas*, *Enhydrobacter*, *Methylobacterium*, *Pseudomonas*, *Ralstonia* spp., *Sphingomonas*, and *Streptococcus*, which may represent novel targets in the development of ophthalmic antimicrobials [109,110,111].

### 3.7. New Approaches for the Prevention and Management of BRDI

New biomaterials based on antibiotics, probiotics, physicochemical coatings, or antibiotic enhancements (ultrasound, bioelectric effect) have been described as effective [112,113]. Some examples of compounds that inhibit biofilm formation are the following: extracellular polymeric substance synthesis inhibitors, adhesion inhibitors, quorum sensing inhibitors, efflux pump inhibitors, cyclic diguanylate inhibitors, nature-derived bioactive scaffolds, antimicrobial peptides, bioactive compounds isolated from fungi, non-proteinogenic amino acids, and antibiotics [114,115,116,117]. Moreover, leading research has been conducted on the effects of phages and their individual proteins on biofilm [118]. Finally, nanotechnology also provides innovative therapeutic approaches to improve the eradication of BRIs [119,120,121,122]. However, the combination of these anti-biofilm agents with antibiotics appears to be overall more effective than treatment with either compound alone [117].

Moreover, according to the treatment procedures to improve biofilm detachment, some research has been described. For example, the use of the Mucus Shaver was demonstrated to be highly effective against biofilm formation on ETTs [123]. In addition, ultrasonic devices or cold atmospheric pressure plasma significantly reduced biofilm on titanium discs to prevent peri-implant infections [124,125].

In the search for future diagnostic procedures, nucleic-acid detection, enhancing culture techniques, using novel microbe imaging, and targeting local immune response should be considered [126]. The latter is considered to play an important role in biofilm development, as a deeper appreciation of host immune system factors is likely critical to understanding and solving the problem of biofilm-related chronic infection [127]. Moreover, approaches for the in vitro and in vivo biofilm models to optimize diagnostic and therapeutic procedures are required [128].

## 4. Concluding Remarks

The presence of biofilm can be linked to the severity and prognosis of both tissue and medical device-related infections. Indeed, BRIs raise the risk of disease and death, as planktonic cells can disseminate from the primary site of infection into the bloodstream, particularly in immunocompromised patients. At the same time, biofilms that adhere to tissues may lead to chronic recurrent infections, requiring, in many cases, surgical treatment. In a BRDI, the main problem is that, usually, the device cannot be removed, making the infection site difficult to treat.

There are major advances in the prevention and treatment of BRIs based on new compounds and technologies that have been shown to be highly effective in in vitro models but require further validation in the clinical setting.

## Figures and Tables

**Figure 1 microorganisms-10-01259-f001:**
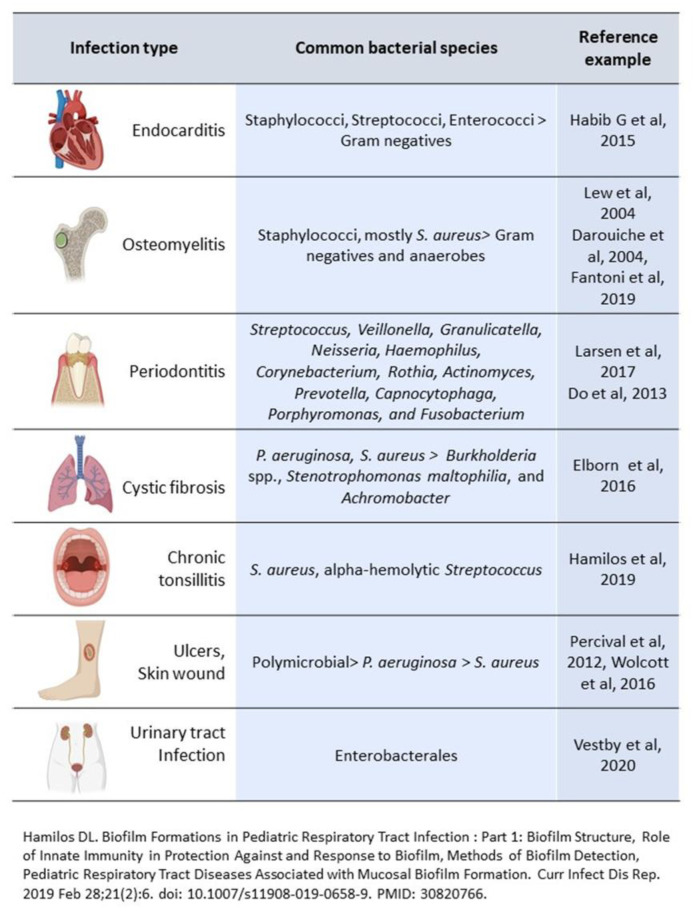
Biofilm and chronic infections in tissue [12,21,22,23,24,25,26,27,28,29,30].

**Figure 2 microorganisms-10-01259-f002:**
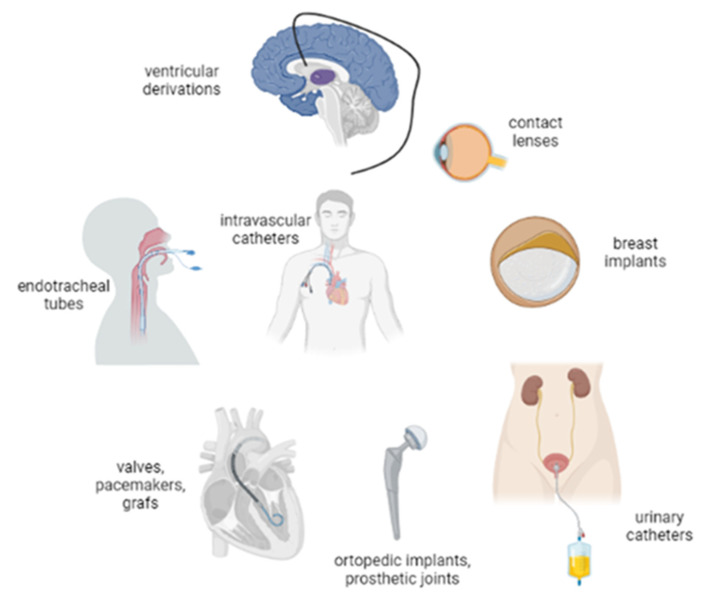
Types of device-related infections.

**Figure 3 microorganisms-10-01259-f003:**
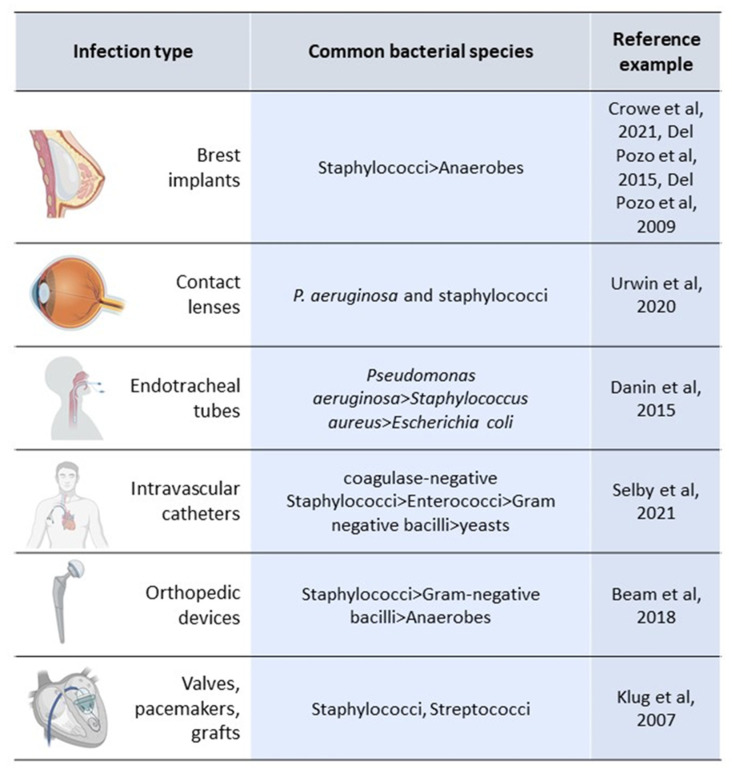
Biofilm and device-related infections [55,56,57,58,59,60,61,62].

## Data Availability

Not applicable.

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
