# Peer review of "The Current Knowledge on the Pathogenesis of Tissue and Medical Device-Related Biofilm Infections"

_microorganisms, 2022, doi:10.3390/microorganisms10071259_

Round 1
Reviewer 1 Report
The Manuscript by Di Domenico et al. is well written and can be accepted for publication in Microorganisms after minor modifications.
Line 73: "human" appears to be written in a different font/size. Please correct.
Author Response
We do appreciate the reviewer's comments. We have checked "human" and it is in a correct font size.
Reviewer 2 Report
A review article “The current knowledge on the pathogenesis of tissue and medical device-related biofilm infections” from Domenico et al explores an interesting topic. But in order to increase the usefulness and significance of the study, it needs a revision before being considered suitable for readers. I think that the paper should be improved in these aspects:
1. To describe and discuss biofilm infection more pictorial and tabular data required, a single figure wouldn't be enough.
2. Author described updates on medical device-related biofilm infections very loosely. So, I will recommend changing the type of manuscript from full length article/review to short communication.
3. Lack of recent reports on pathogenesis of tissue and medical device-related biofilm infections. Authors need to cite recent reports and need to discuss them concisely.
4. Author needs to describe pathogenesis in medical device-related biofilm infections such as contact lens, dental, prosthetic heart valves, and breast implants.
5. Too less discussion about the pathogenesis of tissue and medical device-related biofilm infections. More discussion is needed.
6. Throughout the paper the bacterial species names need to be italicized.
7. Line 73 check font size.
8. It is suggested a moderate English revision by an English native speaker in order to polish text from typos and imperfections.
9. Double check the way of adding references in the main text body. It looks different from the guide of this journal. Additionally, the reference formats do not match this journal.
Author Response
A review article “The current knowledge on the pathogenesis of tissue and medical device-related biofilm infections” from Domenico et al explores an interesting topic. But in order to increase the usefulness and significance of the study, it needs a revision before being considered suitable for readers. I think that the paper should be improved in these aspects:
- To describe and discuss biofilm infection more pictorial and tabular data required, a single figure wouldn't be enough.
We have included new figures (1 and 3) regarding biofilm.
- Author described updates on medical device-related biofilm infections very loosely. So, I will recommend changing the type of manuscript from full length article/review to short communication.
We were encouraged to write a “perspective”, which is described in the journal as: “Perspectives are usually an invited type of article that showcase current developments in a specific field. Emphasis is placed on future directions of the field and on the personal assessment of the author. Comments should be situated in the context of existing literature from the previous 3 years. The structure is similar to a review, with a suggested minimum word count of 3500 words”. The current length of the manuscript is 5265 words and we have taken into account the most recent updates during last years. However, if the editor considers that it should be changed to a short communication, we will accept it.
- Lack of recent reports on pathogenesis of tissue and medical device-related biofilm infections. Authors need to cite recent reports and need to discuss them concisely.
There are currently 83 references included. We would appreciate if the reviewer can be more precise and specify to which recent reports do him/her refers to.
- Author needs to describe pathogenesis in medical device-related biofilm infections such as contact lens, dental, prosthetic heart valves, and breast implants.
We have included some information regarding the pathogenesis of contact lens, dental, prosthetic heart valves and beast implants.
- Too less discussion about the pathogenesis of tissue and medical device-related biofilm infections. More discussion is needed.
We have included some more aspects regarding pathogenesis of tissue and medical device-related biofilm infections.
- Throughout the paper the bacterial species names need to be italicized.
We have checked bacterial species names and all are in italics.
- Line 73 check font size.
Font size has been checked in line 73 and we don’t see any error.
- It is suggested a moderate English revision by an English native speaker in order to polish text from typos and imperfections.
We have done as suggested.
- Double check the way of adding references in the main text body. It looks different from the guide of this journal. Additionally, the reference formats do not match this journal.
We have reformatted references to match this journal.
Round 2
Reviewer 2 Report
The figures are good. The manuscript looks good.